# Building NED: Open Access to Australia's Digital Documentary Heritage

**Barbara Lemon** [1] **, Kerry Blinco** [2] **and Brendan Somes** [3,*]

1    National and State Libraries Australia, Melbourne, Victoria 3000, Australia; blemon@slv.vic.gov.au
2    Library and Archives NT, Darwin City, Northern Territory 0800, Australia; kerry.blinco@nt.gov.au
3    State Library of New South Wales, Sydney, New South Wales 2000, Australia
*    Correspondence: nsla@slv.vic.gov.au; Tel.: +61-3-8664-7512

**Abstract:** This article charts the development of Australia's national edeposit service (NED), from concept to reality. A world-first collaboration between the national, state and territory libraries of Australia, NED was launched in 2019 and transformed our approach to legal deposits in Australia. NED is more than a repository, operating as a national online service for depositing, preserving and accessing Australian electronic publications, with benefits to publishers, libraries and the public alike. This article explains what makes NED unique in the context of global research repository infrastructure, outlining the ways in which NED member libraries worked to balance user needs with technological capacity and the variations within nine sets of legal deposit legislation.

**Keywords:** open repository; legal deposit; Australia; digital heritage; electronic publications

## 1. Introduction

The national edeposit service (NED) was born as Australia's solution to an intractable problem. Prior to 2016, Australia's national legal deposit provisions did not cover electronic publications. Our national, state and territory libraries, working across nine jurisdictions, each boasting vast collections of nineteenth- and twentieth-century print publications, were faced with a proliferation of digital material in increasingly varied formats. Only two of these libraries had the benefit of local legislation that specifically addressed it. Not all had the technical infrastructure or staff capacity to cope with it. Each was attempting to collect the published output of Australia in these differing circumstances with increasing levels of publisher confusion and duplication of holdings [1]. Internal analysis presented to National and State Libraries Australia (NSLA) found in 2014 that approximately 50 per cent of the legal deposit holdings of any one of these nine libraries could be found in another.

In the days of print-only, this degree of duplication was necessary. Australia from east to west is, after all, wider by several hundred kilometres than the diameter of the moon. Requiring publishers to deposit several copies of the same electronic publication in various locations made much less sense.

After a decades-long campaign, Australia's national legal deposit provisions were extended to cover electronic materials through changes to the Copyright Act in 2016 [2]. NSLA members saw an opportunity to build on their long history of collaboration and reduce the complexity of legal deposits nationwide. A new single service would aid publishers in meeting their legal obligations, and could manage at scale the deposit, storage, preservation, discovery and delivery of published electronic material across the country. It would provide a consistent user experience to researchers and members of the public in even the most remote corners of Australia.

NED is certainly not the first national repository for legal deposit publications. Nor is it the first repository that connects multiple libraries with legal deposit mandates [3]. In the United Kingdom, for example, the British Library has been collecting electronic legal deposit publications since legislation

was extended to cover these materials in 2013. Once collected, these publications can also be claimed by the five other legal deposit libraries in the UK. This system is similar to NED in that only one point of deposit is required for publishers—the British Library's Publisher Submission Portal—with customised solutions offered to publishers who produce more than 50 items per year. The British Library has additional capacity to collect published material for legal deposit purposes through web archiving [4].

Where the UK system differs is in its approach to public access. Electronic publications are available only from the reading rooms of legal deposit libraries, discoverable through their catalogues or through the UK Web Archive. In the British Library, a reader pass is required for access. Electronic publications are not made available until at least seven days after collection, and only one reader can access any one publication at the same time.

Similarly, in Germany, where legislation has been in place since 2006, publishers are provided with a number of mechanisms for the deposit of electronic publications and associated metadata. These include web forms, harvesting by arrangement, and deposit into a 'hotfolder' [5]. Records are viewable in the catalogue of the Deutsche Nationalbibliothek, the German National Library, as soon as publications have been transferred to its repository, but public access is onsite only in the library's reading room [6].

A number of factors make NED unique. Firstly, its nine partners co-designed and co-invested in the service, working to meet the requirements of their jurisdictions across a vast geographical distance, with an ongoing operational group comprising representatives from all member libraries. Secondly, it provides very broad public access to publications, with remote access permitted for the majority of content and a minimum of onsite access for all content. NED is exceptional in its provision of a dedicated support service for libraries, publishers and users. It encompasses deposit, collaborative data management, long-term preservation, secure storage, and multi-channel access. It provides a consistent experience for users while preserving local identity and accommodating continuing 'ownership' of publisher relationships by libraries in their jurisdictions. It gives a sense of agency to publishers who can supply their own metadata and nominate access conditions for their works, while facilitating enhancements to item records by member libraries.

*Concept to Reality*

The national, state and territory libraries already had a blueprint for the level of collaboration required to build NED. As members of NSLA, they had been working together in various guises since 1973. In 2007, with the National Library of New Zealand, they established the *Reimagining Libraries* program with a suite of collaborative projects aimed to improve library services and covering everything from copyright policies to accessioning tools and guidelines to working with Indigenous communities [7]. Importantly, each was a contributor to the Trove national discovery service, hosted by the National Library of Australia. It was agreed that content delivery for NED would be managed through Trove, and that NED would be built upon the National Library's existing edeposit system.

Following early scoping and survey work in 2014–2015, there were three major stages leading to the launch of NED in 2019. Stage 1 (2016) saw a steering group established with representation from all nine libraries, and working groups in operation with nominated responsibility for legislation, service modelling, policies and strategies. Stage 2 (2017) included development of detailed technical specifications by the National Library's IT team in close consultation with member libraries, and a formal business case endorsed by NSLA chief executives. User experience design and early publisher communication was undertaken in this stage along with a branding proposal to ensure visibility of all contributors. The NED Deed was finalised, specifying the approach to governance, membership, fees, risk management, publisher liaison, data management, and dispute resolution.

Stage 3 (2017–2019) was the two-year build phase for NED. Here the system was designed, built and tested, including five rounds of collaborative user experience testing, three months' end-to-end and penetration testing, and migration of existing electronic content for some member libraries. The NED website was built and user guides for staff developed, (see NED home page at Figure 1). The NED

operational group was established with system experts in each library, and a NED support officer appointed. Access agreements were obtained from commercial publishers and collective policies for content, publisher management, privacy, and terms and conditions of use completed. NED went live in May 2019 as a 'minimum viable product' (MVP), and was formally launched by the Minister for Communications, Cyber Safety and the Arts in August of that year.

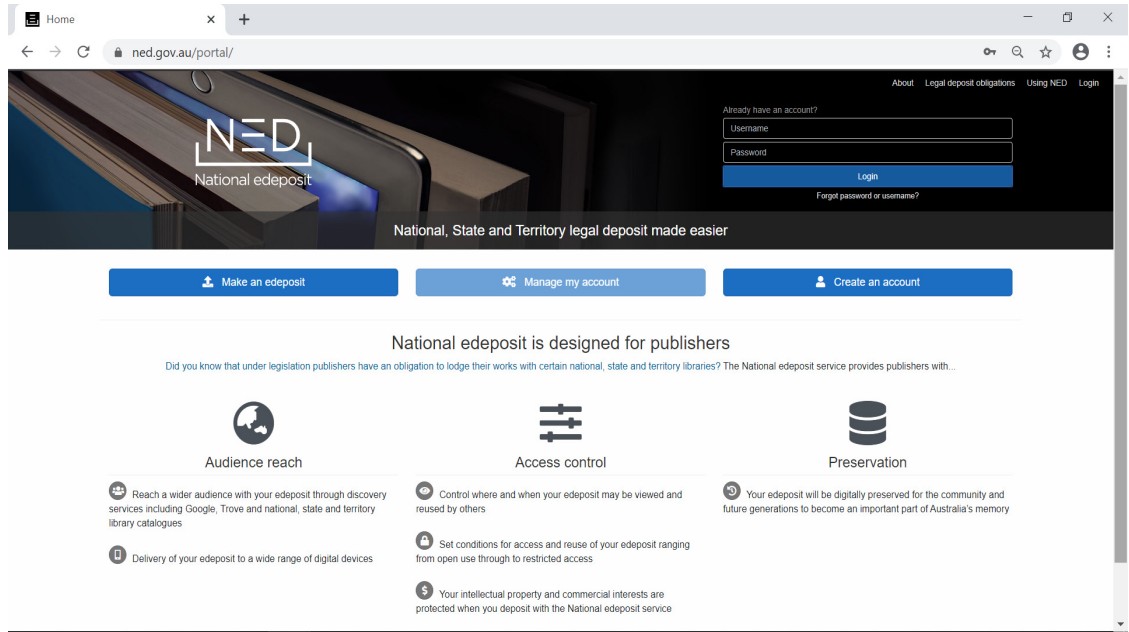

**Figure 1.** Home page of national edeposit service (NED) website and portal, ned.gov.au.

The sections below outline the ways in which NED is designed to meet the user needs of libraries, publishers, and the public, concluding with the current status of NED and planned system and service enhancements.

## 2. National Edeposit Service (NED) for Libraries

The most practical driver for libraries' involvement in NED was the need for efficiency savings in collecting electronic legal deposit publications. Tied to this was a desire for a simplified system for publishers, a more consistent experience for library users, and the satisfying concept of one national system for preservation of and access to the country's digital documentary heritage. For libraries, NED offered the promise of smoother workflows, less duplication of effort, common technical training, improved task allocation, process harmonisation, and reduced costs.

Defining the technical and legislative needs of the collecting institutions partnering in NED was a collaborative and iterative process. From the outset, members worked together to detail the core institutional requirements for the national service as follows:

- support the common intention and purpose of legal deposit legislation across Australia, ensuring the record of published materials within a jurisdiction is comprehensively collected, maintained and preserved for future generations;
- provide a single point of deposit for publishers to meet both national and state/territory legal deposit obligations;
- enable ongoing building and management of publisher/institutional relationships within a jurisdiction;
- provide efficient operational processes across the 'collect, store, manage, provide' lifecycle of legal deposit;

- provide a cost effective solution, building on the existing investment of the National Library of Australia's edeposit service;
- provide branding which identifies the joint service to publishers and staff, and recognises at collection and delivery the responsibilities and contributions of the individual partner institutions;
- operate under a robust governance, decision making and policy framework; and
- include the ability for each institution to identify and/or obtain a copy of the material relating to their jurisdiction.

Each institution needed to evaluate independently the national requirements against their own, and the capacity of the national system to interoperate with local systems and policies.

During Stage 1, high-level requirements were collected through workshops, consultation visits to each of the NSLA libraries and a series of commissioned papers addressing core requirements.

In Stage 2, a business case outlining costings and more detailed requirements was prepared by the steering committee. The technical team at the National Library of Australia was engaged to lead the development of specifications for the system. The specification process was divided into work packages aligned with the legal deposit collecting lifecycle. A wiki was used to distribute discussion papers and draft documents to members of the steering group. Institutions were able to comment on documents, answer questions from the technical team, and vote on preferred options through this platform. The resulting specifications represented a joint understanding between the partners and became part of the NED partnership agreement. Although time-consuming, this collaborative approach had the benefit of creating strong, clear agreement between the partners with a focus on commonalties, a desire to solve problems, and the benefits of NED instead of the differences between jurisdictions.

During Stage 3, institutions were able to add further detail to requirements and provide clarification to the development team through the wiki and during user acceptance testing. It became evident at this time that the audience for formal policies on content, privacy, takedown requests, publisher management, security and access were primarily going to be the partner institutions themselves. Weekly teleconferences and regular face-to-face meetings by the steering group were crucial. Ultimately, constraints around time and budget drove the prioritisation of certain requirements in order to launch the NED service as a minimum viable product.

*System and Operation*

Whilst the most visible component of NED is the user portal, the NED end-to-end service is a complex set of interoperating components located at the National Library of Australia, individual institutions and third-party suppliers as follows:

- **Collecting publications.** This includes the NED publisher portal itself (ned.gov.au), along with three alternative means of collecting: bulk deposit using file transfer protocol (FTP), deposit via member libraries, and direct email subscription for publications such as electronic newsletters.
- **Storage.** This involves digital preservation activity, a central repository for storage of all NED materials, and separate institutional repositories for those member libraries storing copies.
- **Management and description.** This relies on an intersection between the NED administration portal (again accessed via ned.gov.au), member libraries' own library management systems, and the Libraries Australia service (Australia's combined national bibliographic database housed at the National Library).
- **Discovery.** Content held by NED can be accessed through multiple channels: the NED viewer (for those with an account), the Trove discovery service online, Google, WorldCat and individual library catalogues.

Most publishers will interact with NED through the user portal, however the service supports a number of additional workflows, including bulk deposit from large publishers, email subscription to newsletters and collection by partner institutions who then deposit to the national system. Partner

institutions may elect to receive copies of publications deposited in NED to include in their own institutional repositories. Metadata transfers make NED material discoverable in Trove, WorldCat and partner institutional catalogues and discovery services.

Regardless of the method used to collect a publication, the end result is that national and state or territory legal deposit obligations are met with a single deposit, and made available to the public as widely as access conditions allow.

Figure 2 over the page illustrates an eight-step process for data flow. Numbers in the diagram correspond to the steps below:

1. The item is deposited via NED portal, email or FTP for bulk uploads. The publisher chooses one of six access conditions (see next section).
2. The item moves into the central Digital Library Collection (DLC).
3. MARC (machine-readable cataloguing) fields are generated and sent to Libraries Australia [8] via its Record Import Service (RIS).
4. The delegated member library (ML) is notified of the new deposit and its Libraries Australia ID number. The state in which a publisher is based determines the 'owning' library for each item.
5. The ID number is added to the NED system.
6. Libraries Australia data goes into the Trove discovery layer [9].
7. The item goes into the Preservica preservation system and content is backed up offsite (the National Library as host of the NED service securely stores all deposited publications in accordance with government security standards and requirements [10]).
8. As enhancements are made, record data moves from Libraries Australia to and from institutional library management systems (ILMS) and the National Library's Voyager (library management) system, with a separate workflow for the Northern Territory using WorldCat. Based on nominated access conditions, the item is displayed through Trove and NED member library catalogues.

The interaction between systems in Figure 2 is enabled by standards. ONIX (ONline Information eXchange) data can be received from publishers. Metadata collected in NED is profiled and encoded in MARC for transfer to Libraries Australia, from Libraries Australia into WorldCat, and between Libraries Australia and partner institutions' library management systems. Copies are transferred to institutional repositories using Metadata Encoding and Transmission Standard (METS) packages, with MARC XML used for bibliographic data (machine-readable cataloguing in XML, or eXtensible Markup Language), Dublin Core for serials issue data, and METS metadata encoding for rights, access conditions and technical data.

The NED system had to be built by combining and adapting a wide range of existing hardware and software solutions. Content delivery based on Trove is governed by controls employing Citrix, firewalls and Internet Protocol (IP) range restriction. A new web application was built using Jetty for the web server and Java 8 on Spring Boot for core application functionality, Mysql 5.7 for the datastore, and Groovy and Spock for automated tests. The user interface itself used VueJS and javascript for interactions and Bootstrap, CSS and LESS for styling (Cascading Style Sheets and Leaner Style Sheets being languages for describing the presentation of web pages). The National Library's Preservica and Digital Object Storage System were employed for preservation and storage.

Since NED went live, the user needs of libraries have been continually monitored through the NED operational group, comprising one representative from each member library and convened by the NED Support Officer. Deposits can be readily tracked by all NED members, including publishers registered for their jurisdiction, types of deposits made, and status of all deposits. Reporting dashboards are attractive and easy to use, as in Figure 3 below. Figure 4 follows with an expanded view of one data category—'publications by type'.

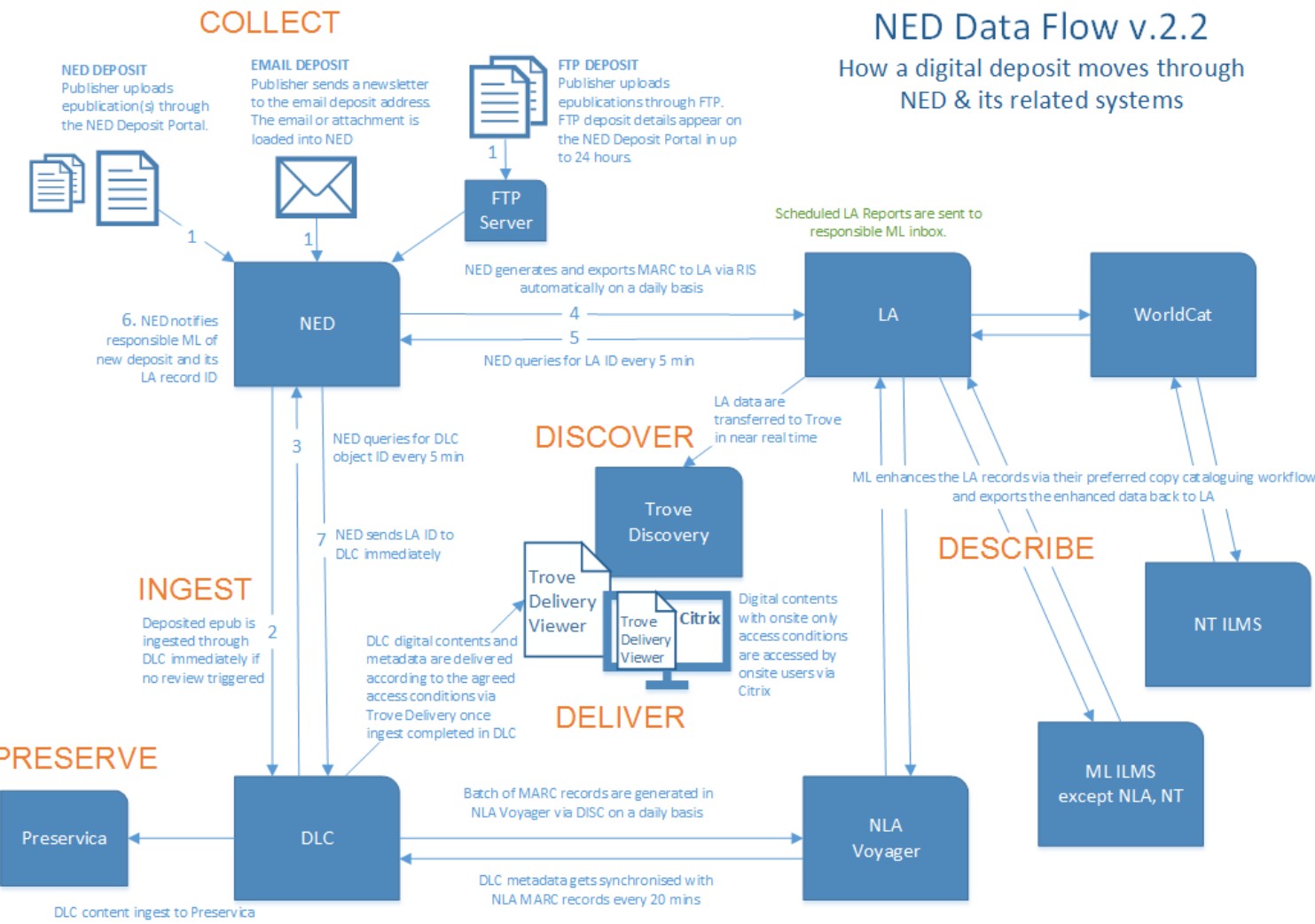

**Figure 2.** Data flow in NED.

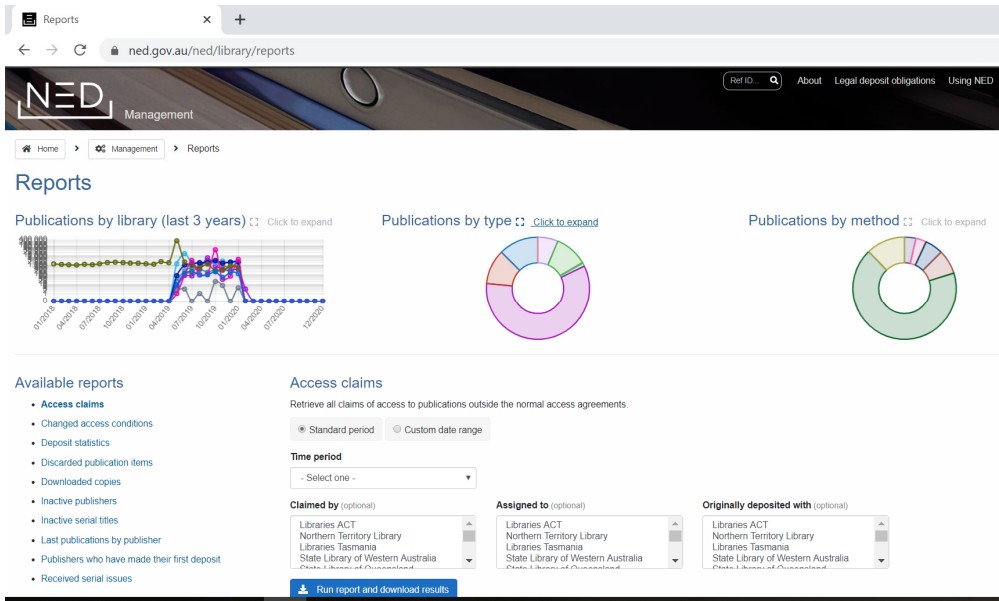

**Figure 3.** Reporting dashboard in NED portal.

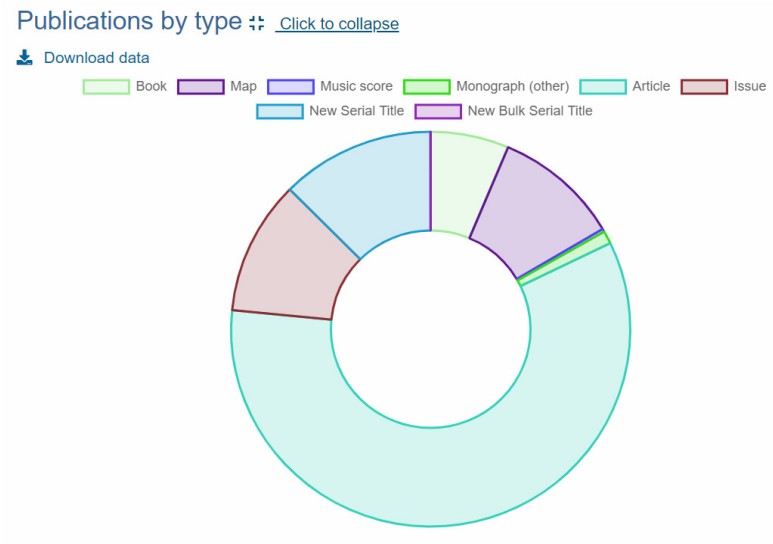

| Type | Number | Percentage |
|---|---|---|
| Book | 28,763 | 6% |
| Map | 48,522 | 11% |
| Music score | 1154 | <0% |
| Monograph (other) | 5377 | 1% |
| Article | 264,126 | 57% |
| Issue | 56,785 | 12% |
| New Serial Title | 56,091 | 12% |
| New Bulk Serial Title | 98 | <0% |
| TOTAL | 460,916 | 100% |

**Figure 4.** Expanded view of 'publications by type' and table of corresponding figures (as at March 2020).

### 3. NED for Publishers

Publisher needs for NED were primarily clarity, simplicity, security and access control. The chief benefit of the NED service for publishers is that it provides a single point of deposit for the fulfilment of national legal deposit obligations [11,12].

However, there are vast differences between organisations defined as 'publishers' of electronic content. They may be commercial or non-profit, publishing one or 1000 publications annually, technologically advanced or lacking resources and capacity in this area. Their specific requirements had to be established by direct consultation. Member libraries already had first-hand experience of publishers depositing digital publications with pre-NED systems, and consulted with individual publishers throughout the development of NED. The Australian Publishers' Association and the Small Press Network were also engaged and regularly consulted throughout.

Many publishers were concerned with security measures, given that publications were to be deposited free of digital rights management and therefore susceptible to copying and distribution. A range of policy and technical solutions were deployed to address these concerns, including functionality allowing publishers to nominate one of six access conditions as follows:

- openly on the internet;
- openly on the internet—with temporary download restriction;
- openly on the internet—with download restrictions;
- openly on the internet—with temporary download and location restrictions;
- onsite-only at national, state and territory libraries; and
- onsite-only at the National Library of Australia and applicable state/territory libraries.

The requirements of the Copyright Act mean that all publications, at a minimum, are made available to view onsite at the relevant deposit library and may be considered for interlibrary loan. Over time it is hoped that the preference for open access will increase as publishers build trust in the service, and recognise its broader benefits, noting that their material will be catalogued, stored, preserved and made accessible now and for all time.

The NED service aims to encourage open access while protecting the commercial viability of publications. Terms and conditions of use were released early to publishing bodies, and access agreements obtained in the case of major commercial publishers. Articles addressing security and access in NED were published in the Australian Publishers' Association newsletter [13], with the NED security and access policy published on the NED website in language that was not library-centric. A particular innovation of the system was the addition of the Citrix viewer [14] as a solution to address publisher concerns around content security for items with commercial value. The viewer restricts functionality to reduce the likelihood of illegal copying or distribution of content.

Of course, the common requirement for all publishers was for NED to be efficient, intuitive and easy to use. The website was carefully designed with this in mind, stepping publishers through an easy-to-use deposit system with an option of depositing once as a guest or setting up an account to track and facilitate more frequent deposits. Metadata requirements were designed so as not to be too onerous and to facilitate reuse from other platforms, such as those used by commercial publishers. A plain language NED user guide and outline of legal deposit requirements is readily accessible from the home page [15].

NED accommodates a range of channels for deposit—including bulk upload and automated subscription to newsletters—and a range of file types to accommodate monographs, serials, music scores and maps. After six months of operation, 8780 publishers were registered with the service.

## 4. NED for the Public

For the Australian public and for overseas researchers seeking access to Australian publications, NED provides one source for Australia's digital documentary heritage, and one consistent viewing experience, but with multiple access points to aid discovery. Publications in NED can be searched within NED member library catalogues, directly from within Trove, or from Google. Those publications with open access conditions can be downloaded and read from wherever the library user happens to be, including from mobile devices.

User experience testing and design was part of Stage 2 for NED, with wireframes shared and concepts tested at a number of conferences and workshops. The Trove-hosted viewers for NED content include features such as searchable text for books and journals, ability to download PDF and JPG files, and deep zoom for pictures and maps. The most restricted items in Trove can be accessed through the Citrix viewer, with downloading and copying disabled (see Figure 5).

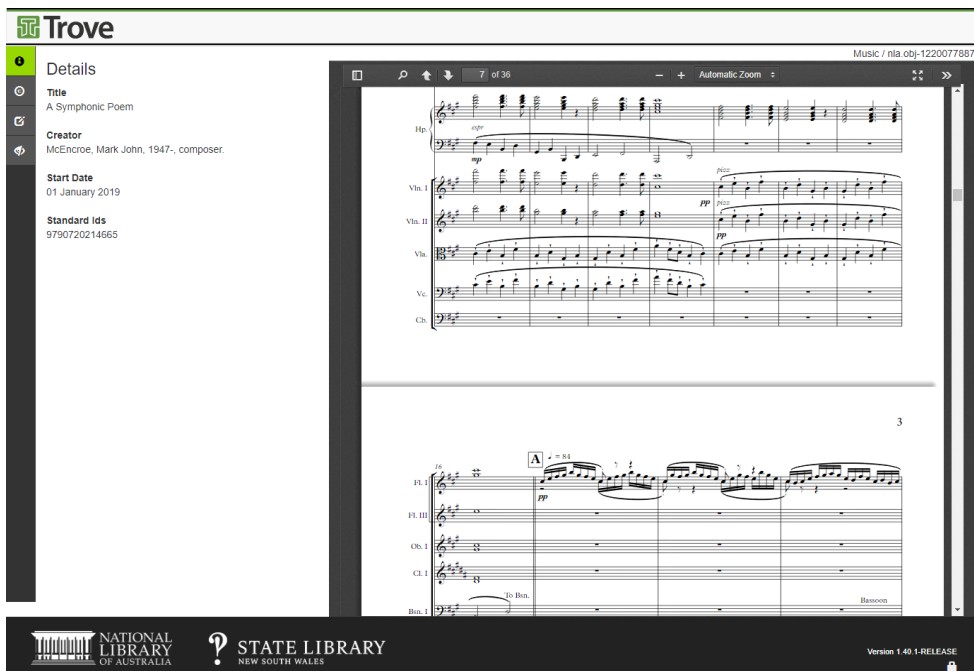

**Figure 5.** Example of a restricted access item in NED, using the Citrix viewer.

Already the range of publications deposited in NED is extraordinary. From bestsellers like *John Curtin's War* (John Edwards, 2017) or *Skin, Kin and Clan: The dynamics of social categories in Indigenous Australia* (McConvell et al., 2018) through to the newsletters of small community organisations such as *Radio-gram: news from the Vintage Wireless and Gramophone Club* in Perth or *Christian Biker* from New South Wales. Multilingual holdings will be a growth area for NED, building on existing holdings such as *The Australian panorama Arabic newspaper*.

## 5. Next Steps: Beyond MVP

Three months after launch, NED had received nearly 7000 new deposits from publishers and approximately 110,000 existing edeposit works had been migrated from member library systems. The service was operating as 'business as usual' in member libraries, anticipating higher and higher numbers of deposits as publishers grew familiar with the portal and as new publishers of digital content received communications about their legal deposit obligations.

NED was launched as a minimum viable product, however. At the time of launch in August 2019, a total of 99 outstanding issues had been identified by the NED operational group. A large number of these were minor but approximately twenty were classified as requiring substantial work by the IT

team for resolution. Examples of minor issues were incorrect field displays, permissions settings for library staff, or summary tables for publishers not reflecting the ISSNs for their serial deposits. Larger issues requiring detailed scoping and investment include:

- conversion from monographs to serials, based on a regular publisher error of depositing serial issues as if they were monographs;
- problems with bulk FTP functionality for major publishers, required in the transition from print to digital for commercial serial publishers, including large media companies;
- automation of serials claiming processes, so that new editions can quickly be added to a series without re-entering metadata;
- improved functionality for hierarchy and multi-volume items;
- improved integration of publisher metadata; and
- integration of non-NSLA legal deposit libraries in Queensland, New South Wales and South Australia. Each of these three states have parliamentary libraries included in their state legal deposit legislation, with the addition of two university libraries for Sydney.

Further challenges have been encountered in the migration of existing edeposit material from member libraries, given the duplication in collections and the wide range of systems and standards that have been in use since electronic materials were first collected.

Ascertaining whether the service is meeting requirements identified by the NED operational group will be possible with data analysis generated by Libraries Australia and the NED system, based on coding in the NED item records. NED will allow us, for the first time, to measure nationwide access and use of legal deposit collections and to gauge the level of take-up of the service by publishers. Publisher concerns or queries will be readily addressed by their assigned library (based on place of publication) or by NED support.

NED will remain a priority project for NSLA libraries with continuing full-time development work funded. In future, it is hoped that the service can evolve to incorporate a wider range of formats and meet the changing needs of its various user groups, forming part of suite of collaborative systems and services for NSLA libraries including web archiving and audio-visual material.

**Author Contributions:** Conceptualization, B.L.; writing—original draft preparation, B.L., K.B., B.S.; writing—review and editing, B.L. All authors have read and agreed to the published version of the manuscript.

**Funding:** This research received no external funding. It was jointly funded by the national, state and territory libraries of Australia.

**Acknowledgments:** For early concept work and scoping, Warwick Cathro, Alison Dellit, Marie-Louise Ayres, Kate Irvine, Anna Raunik and Alison Sutherland. The authors wish to acknowledge the considerable work of members of the National Library of Australia's digital team, with members of the NED steering group and NED operational group in launching the NED service. We commend the directors and chief executive officers of all nine NSLA libraries for their collaboration and investment in NED.

**Conflicts of Interest:** The authors declare no conflict of interest.

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
