# Peer review of "Building NED: Open Access to Australia’s Digital Documentary Heritage"

_publications, doi:10.3390/publications8020019_

Round 1

Reviewer 1 Report

I have no concerns about the content but Figures, I thinkk figures 1, 4 and 5 could be removed from teh text they are just screenshots that ay user can visualize in the portal

Author Response

Point 1: I have no concerns about the content but I think Figures 1, 4 and 5 could be removed from the text. They are just screenshots that any user can visualise in the portal.

Response to point 1: Respectfully, we have opted to leave these images in the text as the focus of the article (and this edition of the journal) is the user experience – the ways in which repositories have responded to user needs. In this instance we feel that a visual depiction of that experience is just as important as a written description of it. We are happy for this decision to be overridden on final review.

Reviewer 2 Report

This is  an interesting paper about a big project of digitalization of australian heritage. It is a clear and detailed explanation of the process accomplished, including all the stakeholders who have participated. 

Maybe it could be improved the graphic of publications (line 205-206), including percentages in numbers. 

Overall it is a very good explanation of the project that could be replicated by other countries.

Author Response

Point 1: Maybe it could be improved the graphic of publications (line 205-206), including percentages in numbers.

Response to point 1: Percentages and numbers have been added beneath the graphic, separated out in Figure 4 at line 265. Numbering for other figures has been updated throughout the document.

(Please note that line numbers provided correspond with the document only when viewed in ‘Simple Markup’ mode.)

Reviewer 3 Report

I would like to draw attention to figures 2 and 3.

Figure 2: the little images do not really illustrate but rather make the concept harder to grasp. 

Figure 3: for a figure, it is loaded with text, including abbreviations like ML, LA, DLC, RIS, ILMS that do not seem to get explained. Much is expected from readers regarding their eyesight, deductions skills, and knowledge of library jargon. Insofar this will be read online, the orientation is awkward, too.

For the rest, the article is interesting in its subject matter, and makes an enjoyable read. 

"Peak body" seems to be an Australian expression that one gets after a little confusion. 

Author Response

Point 1: Figure 2: The little images do not really illustrate but rather make the concept harder to grasp.

Response to point 1: We have replaced the graphic with a written explanation at lines 178-190.

Point 2: Figure 3: for a figure, it is loaded with text, including abbreviations like ML, LA, DLC, RIS, ILMS that do not seem to get explained. Much is expected from readers regarding their eyesight, deductions skills, and knowledge of library jargon. Insofar this will be read online, the orientation is awkward, too.

Response to point 2: The figure has been designed by system developers in a partner institution using design software in a format that we cannot now modify. Agreed solution in correspondence with the editor was to add a fuller explanation of abbreviations and content. This can be found from lines 206-223, and the figure has been expanded to a full page to improve legibility.

Point 3: "Peak body" seems to be an Australian expression that one gets after a little confusion. 

Response to point 3: Line 29, deleted “the peak body”.

(Please note that line numbers provided correspond with the document only when viewed in ‘Simple Markup’ mode.)

Reviewer 4 Report

The main contribution of the paper is to present the development of Australia’s national deposit service of electronic publications. Legal deposit of electronic publications is recent as could be argued by mentioning recent literature (e.g. https://www.loc.gov/law/help/digital-legal-deposit/digital-legal-deposit.pdf).

It could be interesting to make an introductory context defining the problem and other countries approach it.

Because of the nature of the paper, a description of a project, most of the categories regarding a scientific contribution is not applicable.

Author Response

Point 1: It could be interesting to make an introductory context defining the problem and other countries approach it.

Response to point 1: We have added further contextual information using the British and German approaches to electronic legal deposit as examples. The paragraph formerly at line 78-90 was moved to line 44 and further information is displayed in tracked changes between lines 45-65. Footnotes have been added (numbers 2-5) and footnote numbering updated throughout article accordingly.

(Please note that line numbers provided correspond with the document only when viewed in ‘Simple Markup’ mode.)